# The Ratio of Nasal Cannula Gas Flow to Patient Inspiratory Flow on Trans-nasal Pulmonary Aerosol Delivery for Adults: An in Vitro Study

**DOI:** 10.3390/pharmaceutics11050225

**Published:** 2019-05-10

**Authors:** Jie Li, Lingyue Gong, James B. Fink

**Affiliations:** 1Department of Cardiopulmonary Sciences, Division of Respiratory Care, Rush University Medical Center, Chicago, IL 60130, USA; Lingyue_Gong@rush.edu (L.G.); fink.jim@gmail.com (J.B.F.); 2Aerogen Pharma Corp, San Mateo, CA 94402, USA

**Keywords:** oxygen inhalation therapy, high-flow nasal cannula, aerosol, flow

## Abstract

Trans-nasal aerosol deposition during distressed breathing is higher than quiet breathing, and decreases as administered gas flow increases. We hypothesize that inhaled dose is related to the ratio of gas flow to patient inspiratory flow (GF:IF). An adult manikin (Laerdal) with a collecting filter placed at trachea was connected to a dual-chamber model lung, which was driven by a ventilator to simulate quiet and distressed breathing with different inspiratory flows. Gas flow was set at 5, 10, 20, 40 and 60 L/min. Albuterol (2.5mg in 1 mL) was nebulized by vibrating mesh nebulizer at the inlet of humidifier at 37 °C for each condition (*n* = 3). Drug was eluted from the filter and assayed with UV spectrophotometry (276 nm). GF:IF was the primary predictor of inhaled dose (*p* < 0.001). When the ratio was < 1.0, the inhaled dose was higher than ratio > 1.0 (21.8 ± 3.8% vs. 9.0 ± 3.7%, *p* < 0.001), and the inhaled dose was similar between quiet and distressed breathing (22.3 ± 5.0% vs. 21.3 ± 2.7%, *p* = 0.379). During trans-nasal aerosol delivery, GF:IF primarily affected the inhaled dose. Compared to the ratio above 1.0, the ratio below 1.0 produced a higher and more-consistent inhaled dose.

## 1. Introduction

High-flow nasal cannula (HFNC) is primarily a method of oxygen administration, in which gas flow exceeds patient inspiratory flow [1]. Incorporation of nebulizers into HFNC delivery systems has gained interest in recent years [2,3,4,5,6,7,8], for its combined benefits of comfort associated with nasal interface, trans-nasal pulmonary delivery of aerosolized medication, and mechanical benefits of HFNC. Aerosol with HFNC has been described with bronchodilators for asthmatic [2], bronchiolitis [3,4], or chronic obstructive pulmonary disease (COPD) patients [5,6,7], and inhaled epoprostenol for patients with pulmonary hypertension or hypoxemia [8]. Interruption of HFNC for administration of standard aerosol treatment for periods up to 15 min may compromise oxygenation, and administration by mouth, while receiving HFNC reduces the inhaled dose. Consequently, aerosol administration via HFNC for short duration of standard aerosol treatment might be beneficial. For patients who need long durations of aerosol administration, such as inhaled epoprostenol for pulmonary hypertension [8] or bronchodilator for refractory asthmatics [2], the use of traditional interfaces such as mask or mouth piece is complicated by lack of patient comfort and tolerance. HFNC has been described as a feasible route to deliver continuous aerosolized medication; clinical observations report that pediatric patients appear more comfortable and less anxious during bronchodilator nebulization via HFNC than mask or mouthpiece [3,4].

Trans-nasal pulmonary aerosol treatment using an HFNC device set up has been described as administering gas flow from 5 L/min to 60 L/min for adult population [5,6,7,8], to support three basic patient populations: 1) Patients with severe hypoxemia who require high gas flow rate and fraction of inspired oxygen (F_I_O_2_) to meet their oxygenation and ventilatory requirements [8]; 2) patients with mild to moderate hypoxemia who require moderate gas flow and F_I_O_2_ [6,7]; and 3) patients whose oxygenation or ventilation is satisfactory but benefit from the nasal interface to inhale aerosolized medication for an extended period of time [7]. Patient inspiratory flow varies in these three populations. In the absence of guidelines on setting or adjusting gas flow during HFNC treatment, the chosen flow can be arbitrary, exceeding or underserving actual patient inspiratory flow.

A radiolabeled in vivo study reports that aerosol lung deposition via HFNC is only 3.6% in adult healthy volunteers [9]. Eight in vitro investigations of influential factors, such as delivery gas type and flow rate, nebulizer type and placement, breathing pattern, size of nasal cannula and role of heated humidification [10,11,12,13,14,15,16,17,18], report that nasal cannula gas flow plays a key role in aerosol delivery via HFNC [10,15,16,18]. In addition, both in vivo and in vitro studies demonstrate that aerosol deposition is inversely related to the nasal cannula gas flow during quiet breathing [10,15,16,18,19]. However, this phenomenon does not appear to extend to distressed breathing, as Dailey et al. reported that aerosol deposition peaked at 30 L/min among gas flows of 10, 30 and 50 L/min [16]. Dr. Ari in her recently published review speculated that aerosol delivery would be maximized when nasal cannula gas flow matches patient inspiratory flow [20]. In contrast, we speculate that aerosol delivery would be maximized when the gas flow is lower than patient inspiratory flow; as aerosol medication is delivered with gas flow that incrementally exceeds the patient inspiratory flow, medication will be increasingly wasted without being inhaled. Conversely, if a patient inspiratory flow increases, wastage will decrease. Moreover, we aimed to further quantify the relationship between the inhaled dose and the ratio of nasal cannula gas flow to patient inspiratory flow (GF:IF) across quiet and distressed breathing.

## 2. Materials and Methods

### 2.1. Experiment Set Up

All experiments were run in a simulated adult model (adult airway management trainer, Laerdal Medical AS, Stavanger, Norway), with size appropriate upper airway anatomy (Figure 1). The mouth was sealed by tape to simulate nose breathing with mouth closed. A collecting filter (Respirgard 303, CareFusion, San Diego, CA, USA) was connected between the distal end of the manikin’s trachea and one chamber of a dual chamber model lung (TTL, Michigan Instruments, Grand Rapids, MI, USA). A rigid bar was connected to a second chamber, which was attached to a critical care ventilator (PB 840, Medtronic, Minneapolis, MN, USA). When the ventilator delivered the preset volume and flow to the chamber, it produced a negative pressure in the chamber, simulating an inspiratory effort. Per manufacture’s recommendations that nasal cannula size be smaller than 50% of the diameter of patient’s nostrils, a large size adult nasal cannula (Optiflow^TM^, Fisher & Paykel, Auckland, New Zealand) was placed on the manikin’s nares, and connected to an adult HFNC breathing circuit (RT202, Fisher & Paykel, Auckland, New Zealand) with a high flow device (Optiflow^TM^, Fisher & Paykel 850 system, Auckland, New Zealand); temperature was set at 37 °C. A mass flowmeter (4040, TSI, Shoreview, MN, USA) was used to confirm flow settings (5, 10, 20, 40 and 60 L/min). A mesh vibrating nebulizer (Aerogen Solo, Aerogen Ltd., Galway, Ireland), with a residual drug volume < 0.1 mL was placed via a T-piece at the dry side (inlet) of the humidifier.

A respiratory profile monitor (NICO2, Respironics, Murrysville, PA, USA; not shown) was connected between the manikin’s trachea and the chamber, to measure the tidal volume and inspiratory flow generated by the manikin.

### 2.2. Comparison between Groups

Ventilator settings in volume control mode with square flow waveform were adjusted to achieve the desired manikin’s breathing profiles (Table 1). During quiet breathing, respiratory rates (*RR*) were set at 15 bpm with inspiratory to expiratory time ratio (*I:E*) at 1:2, three inspiratory flows (13.5, 22.5, 31.5 L/min) were utilized to generate three tidal volumes (*Vts*) (300, 500 and 700 mL). During distressed breathing, respiratory rates were set at 30 bpm with I:E at 1:1, simulated patient inspiratory flows were set at 27 and 42 L/min to achieve *Vt* at 450 and 700 mL. To further evaluate the effect of inspiratory flow on aerosol delivery, we added an experiment with faster inspiratory flow of 52.5 L/min with *Vt* 700 mL, *RR* 30 bpm and *I:E* = 1:1.5.

To compare quiet vs. distressed breathing, we used the settings that were utilized in previously published studies [15,16] (Case 1: *RR* 15 bpm, *I:E* 1:2, *Vt* 500 mL vs. *RR* 30 bpm, *I:E* 1:1, *Vt* 700 mL). To simulate the clinical setting for subjects with small *Vt*, we compared one more case with small *Vt* setting (Case 2: *RR* 15 bpm, *I:E* 1:2, *Vt* 300 mL vs. *RR* 30 bpm, *I:E* 1:1, *Vt* 450 mL).

Five nasal cannula gas flows (5, 10, 20, 40, and 60 L/min) were tested with each breathing profile and three runs were repeated in each condition (*n* = 3). Albuterol powder (1 g, Sigma-Aldrich, St. Louis, MO, USA) was diluted into 400 mL sterile water to prepare the concentration at 2.5 mg/mL, with 1 mL placed in the reservoir of the nebulizer; and operated until no aerosol was generated. Nebulization took 2–4 min to complete. After nebulization, the collecting filter was removed and eluted with 10 mL solution (0.1M HCl mixed with 20% ethanol) and assayed with UV spectrophotometry (276 nm).

### 2.3. Statistical Analysis

The amount of medication deposited on the collecting filter was positioned at the carina of the manikin’s trachea, namely inhaled dose, reflecting the mass of aerosol potentially reaching the lung and calculated as a percentage of the nominal dose (2.5 mg) and expressed as mean ± SD for each experiment. The differences of inhaled dose among five nasal cannula gas flows with each inspiratory flow were analyzed with Friedman test. The differences of inhaled dose among inspiratory flows at each nasal cannula gas flow were compared using the Kruskal-Wallis test. For comparison of the inhaled dose with quiet vs. distressed breathing, the Mann Whitney test was used.

To visualize the relationship between nasal cannula gas flow and patient inspiratory flow with the inhaled dose, a 3D plot was drawn and response surface regression was utilized to report the predicted response surface using SAS statistical software (SAS Institute Inc; Cary, NC, USA). The response surface regression used the method of least squares to fit quadratic response surface regression models. To investigate how these factors predicted inhaled dose, a multiple linear regression was used to assess the 90 depositions. The bivariate relationship between the inhaled dose and the influential factors was investigated using Pearson’s correlation analysis. ANOVA was used to assess the difference of the inhaled dose and patient inspiratory flows and nasal cannula gas flows. Based on results from exploratory data analysis, the variables with *p*-value < 0.20 were entered into a stepwise regression model. A *p*-value of < 0.05 was considered to be statistically significant for all predictor variables. Data analysis was conducted with SPSS statistical software (SPSS 23.0 for Windows; SPSS; Chicago, IL, USA).

## 3. Results

### 3.1. Inhaled Dose during Quiet and Distressed Breathing

During quiet breathing (*RR* = 15 bpm, *I:E* = 1:2, *Ti* = 1.33s), inhaled dose increased as nasal cannula gas flow decreased (*p* < 0.001), with greatest deposition at the lowest flow of 5 L/min. With inspiratory flows of 13.5, 22.5 and 31.5 L/min, the inhaled dose increased as the inspiratory flow increased at all nasal cannula gas flow settings (*p* = 0.027) (Figure 2a).

During distressed breathing, the maximal inhaled dose generated by different nasal cannula gas flow in the three breathing profiles tested. For inspiratory flow at 27 L/min (*Vt* 450 mL, *RR* = 30, *I:E* = 1:1), the inhaled dose was greatest with nasal cannula gas flow of 10 L/min (20.8 ± 0.7%). In contrast, nasal cannula gas flow of 20 L/min produced the peak inhaled dose of 22.0 ± 0.6% and 26.7 ± 0.7%, respectively, for inspiratory flow at 42 L/min (*Vt* 700mL, *RR* = 30, *I:E*= 1:1) and 52.5 L/min (*Vt* 700 mL, *RR* = 30, *I:E*= 1:1.5) (Figure 2b). Inhaled dose was greater with higher inspiratory flows (52.5 and 42 L/min) at all gas flows with the exception of 5 L/min. With inspiratory flow ≤ 20 L/min, the inhaled dose was similar with nasal cannula gas flows of 20, 10 and 5 L/min (*p* = 0.18), but higher than flows at 40 and 60 L/min (*p* = 0.001).

### 3.2. The Impact of Ratio of Nasal Cannula Gas Flow to Patient Inspiratory Flow on Inhaled Dose

The relationship between the inhaled dose and two variables (nasal cannula gas flows and patient inspiratory flows) was graphically represented by a 3D response surface (Figure 3), illustrating the interaction between patient inspiratory flow (X axis) and nasal cannula gas flow (Y axis) on the predicted inhaled dose (Z axis). The predicted inhaled dose increased as nasal cannula gas flow decreased and patient inspiratory flow increased.

A scatterplot was drawn to further explore the relationship between the inhaled dose and the GF: IF ratio. The inhaled dose decreased as the GF: IF ratio decreased (Figure 4a). Using the GF:IF ratio = 1 as a delineator, the inhaled dose was higher with GF:IF < 1, than GF:IF > 1 (21.8 ± 3.8% vs. 9.0 ± 3.7%, *p* < 0.001). When GF:IF was < 1, the inhaled dose was consistent with quiet and distressed breathing (21.3 ± 2.7% vs. 22.3 ± 5.0%, *p* = 0.379). In contrast, when GF:IF was > 1, the inhaled dose was higher in distressed breathing than quiet breathing (11.6 ± 3.1% vs. 7.5 ± 3.2%, *p* = 0.001). Moreover, the inhaled dose was consistent across gas flows in both groups of GF:IF < 1 and GF:IF > 1 (Figure 4b).

Additional scatterplots were drawn to further explore the inhaled dose with GF:IF < 1 (Figure 5a). Using GF:IF = 0.5 as a delineator, the inhaled dose reached a plateau with GF: IF of 0.1–0.5 (Figure 5b) and the inhaled dose was higher than that of GF:IF between 0.51 to 1.0 (23.4 ± 3.3% vs. 18.3 ± 2.2%, *p* < 0.001).

### 3.3. Predictor of Inhaled Dose during Trans-Nasal Pulmonary Aerosol Delivery

The variables of tidal volume, breathing pattern and GF:IF ratio were marginally associated with inhaled dose (*p* < 0.20). All of the three variables were entered into the stepwise regression model to predict the inhaled dose. The GF:IF ratio was the primary independent predictor of increased aerosol delivery (*p* < 0.001). The regression model explained 78.8% of total variance in inhaled dose delivered via nasal route. The regression model for inhaled dose was *Y* = 24.99 – 6.20 × (GF:IF). There was no significant collinearity observed (tolerance > 0.5).

## 4. Discussion

### 4.1. The Ratio of Nasal Cannula Gas Flow to Patient Inspiratory Flow in Trans-Nasal Aerosol Delivery

To our knowledge, this is the first study to comprehensively investigate the impact of the ratio of nasal cannula gas flow to patient inspiratory flow (GF:IF) on trans-nasal aerosol delivery in an adult model. The flow ratio was identified as a primary independent predictor of inhaled dose, playing a more important role in trans-nasal aerosol delivery than nasal cannula gas flow, patient inspiratory flow, and quiet or distressed breathing pattern. To date, four in vitro studies investigated the influential factors in trans-nasal aerosol delivery in an adult model [11,13,15,16]. However, two of the studies were limited to one nasal cannula gas flow and one breathing profile [11,13], which could not provide comparative information. In contrast, the other two studies reported three nasal cannula gas flows in both quiet and distressed breathing patterns [15,16]. In our calculation of the six flow ratios reported by Reminiac et al., we identified that the inhaled dose increased as the flow ratio decreased, consistent with our findings. However, the lowest flow ratio in their study was 0.67, representing the sole flow ratio below 1.0 [15]. In contrast, we delineated 30 flow ratios from 0.10 to 4.44, representing the likely range of flow ratios for adult patients, to confirm our hypothesis.

In a study by Dailey et al., aerosol deposition with a ratio of 0.22 was lower than that of 0.67 in distressed breathing [16], which seems contradictory to the findings of both Reminiac et al. [15] and ours. This difference might be due to placement of the collection filter distal to the nasal prongs [16] instead of the trachea [15]. The anatomic dead space from the upper airway used in both Reminiac et al. [15] and our study may be more relevant to the realistic clinical scenario.

As the definition of high flow oxygen is the administration of gas flow meeting or exceeding patient inspiratory flow, we chose the GF:IF ratio of 1 as the delineator to explore the impact of the GF:IF ratio on inhaled dose with trans-nasal aerosol delivery. We found that the inhaled dose was higher with GF:IF < 1 than that with GF:IF > 1; this supports our hypothesis and agrees with the finding in our pediatric in vitro study [18]. This might be explained that when the nasal cannula gas flow is less than patient inspiratory flow demand, less turbulence flow would be created in the patient’s respiratory tract, and more importantly, a greater proportion of the medication carried by nasal cannula gas would be inhaled by the patient with less medication wasted.

In contrast to the adult model where inhaled dose was greatest and plateaued at the lowest GF:IF ratio of 0.1–0.5, the most efficient delivery with the pediatric model was with GF:IF of 0.28–0.57, rather than the lowest ratio of 0.1-0.27 [18]. This might be a consequence of the greater transit time of aerosol between the nebulizer and the nasal prongs in the pediatric model at the extreme low gas flow rate where residence time of aerosol is greatest, allowing more sedimentation loss of particles. When aerosol is transported by the heated gas with high absolute humidity, hygroscopic growth of aerosol is described, likely resulting in increased deposition in the nasal cannula and circuits [21]. At the low gas flow rate, residence time increases in the conducting circuit, increasing contact time of aerosol particle with the water vapor. However, it is unclear whether increased particle size in a high absolute humidity environment occurs rapidly or over a time with extended exposure. Both Réminiac et al. and Dailey et al. found distressed breathing had greater aerosol deposition than quiet breathing when nasal cannula gas flow was ≥ 30 L/min [15,16], in which calculated GF:IF was above 1 from both experimental settings, with the exception of ratio = 0.67 with gas flow = 30 L/min in distressed breathing in Reminiac’s study [15]. This agrees with our finding that an inhaled dose with distressed breathing was higher than quiet breathing when GF:IF was > 1. However, we observed that inhaled dose became more consistent when GF:IF was < 1, especially with GF:IF as low as 0.1–0.5; the inhaled dose was consistently maintained at > 20%, regardless of patient breathing pattern.

### 4.2. Clinical Implication

The greater the nasal gas flow above patient inspiratory flow, the lower the inhaled dose. Patients with severe hypoxemia and distressed breathing require sufficient nasal cannula gas flow to meet or exceed their inspiratory flow to avoid air entrainment and F_I_O_2_ reduction [1]. In many ICUs, HFNC is often administered at arbitrary levels of 50–60 L/min, which may exceed patient inspiratory flow. When these patients require inhaled medication via nasal cannula, whether for short or extended durations, reducing the gas flow to match their actual inspiratory demand, to a ratio of 1, has the potential to substantially improve aerosol delivery efficiency without the risk of compromising oxygenation. If a patient cannot tolerate a reduced flow, compensation with larger nominal doses may be appropriate.

For those patients who do not require “high” gas flow for oxygenation and ventilatory support, the nasal cannula interface with humidified gas can be a vehicle to administer continuous aerosolized medication for extended periods of time [22]. Nasal cannula is easier to tolerate than masks or mouthpieces for extended durations. For this application, lower nasal cannula gas flow settings administered via HFNC device setups can deliver more aerosols [19], potentially eliciting better clinical response and reducing the nominal dosage of medication required.

The theoretical clinical implication is that one should tailor the nasal gas flow based on patient inspiratory flow to achieve the optimal ratio for aerosol delivery efficiency. It may be complicated currently to measure patient inspiratory flow, but one could envision the development of such systems.

### 4.3. Limitation

In our model, we only investigated the inhaled dose with nose breathing, and our findings do not reflect the inhaled dose via trans-nasal pulmonary delivery for patients who breathe via the mouth. Réminiac et al. found that an inhaled dose was less with mouth breathing than nose breathing at nasal cannula gas flow ≥ 30 L/min [15]; however, whether the finding still exists with nasal cannula gas flow below 30 L/min is still unknown. Future studies might be needed to confirm this. Moreover, our simulator generated a modified square wave inspiratory pattern, which was not strictly a sinusoidal breathing pattern commonly produced by humans. When we compared the modified square wave produced by our current setup with a sinusoidal pattern produced by piston pump (Harvard Apparatus), the inhaled doses were similar.

Similar to other in vitro studies [10,11,12,13,14,15,16,17,18], the findings are based on a limited set of breathing parameters, which do not represent all patients. In addition, our manikin does not have anatomical and physiological structures of the upper airway such as turbinate and airway cilia-mucosa, which may affect the transit of aerosol. Collection filters capture aerosol from gas passing through to our model lung and do not allow the aerosol to be exhaled as seen in vivo. Both of these factors may result in higher deposition numbers in vitro than in vivo.

Clinically, distressed breathing can be associated with increased mucus ciliary clearance, which will potentially affect the model predictions on lung delivery. Future in vivo studies (radiolabeled aerosol inhalation and PK/PD studies) are necessary to confirm our findings and assess the benefits of using the GF:IF ratio. Further clinical studies with careful titration of nasal cannula flow on individual responses are also needed.

## 5. Conclusions

During trans-nasal aerosol delivery via “high-flow nasal cannula” set up, the ratio of nasal cannula gas flow to patient inspiratory flow was identified as a primary independent predictor of inhaled dose. When the ratio was < 1, the inhaled dose was higher than that with ratio > 1. The inhaled dose was also more consistent with quiet and distressed breathing with ratio < 1.

## Figures and Tables

**Figure 1 pharmaceutics-11-00225-f001:**
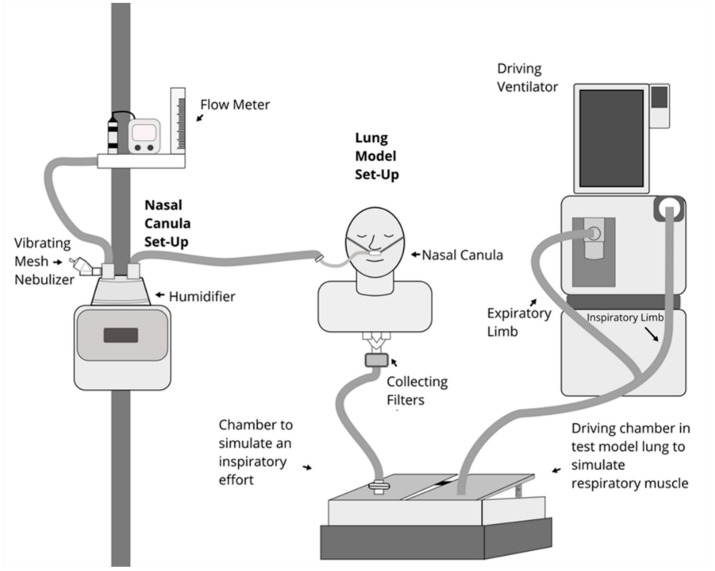
Experiment set up.

**Figure 2 pharmaceutics-11-00225-f002:**
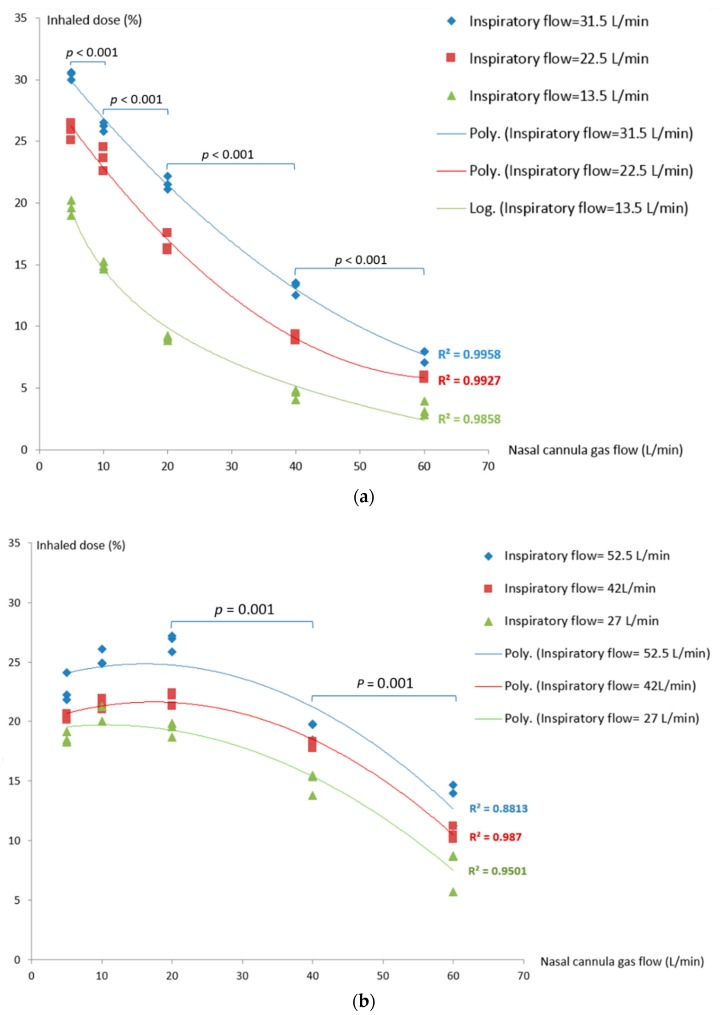
Inhaled dose during different breathing patterns: (**a**) Quiet breathing; (**b**) distressed breathing.

**Figure 3 pharmaceutics-11-00225-f003:**
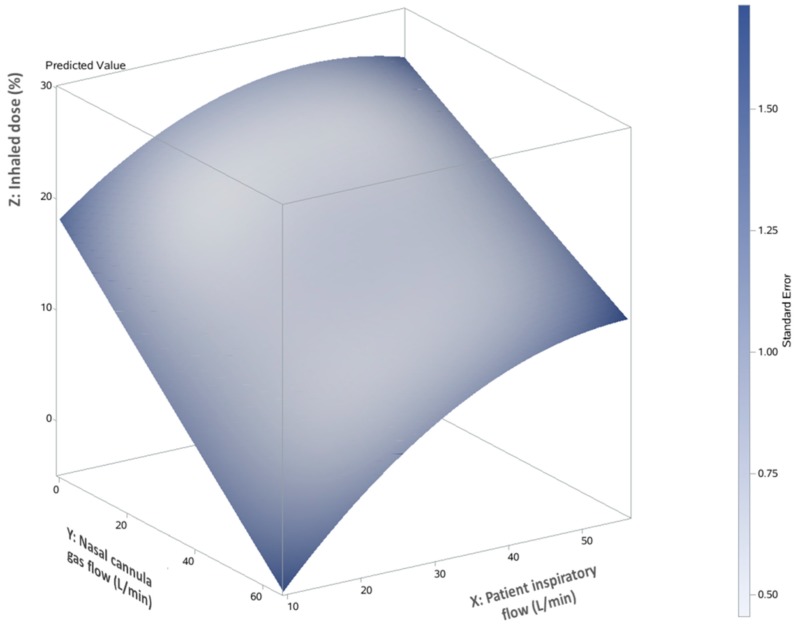
3D response surface and predicted inhaled dose with nasal cannula flow and patient inspiratory flow.

**Figure 4 pharmaceutics-11-00225-f004:**
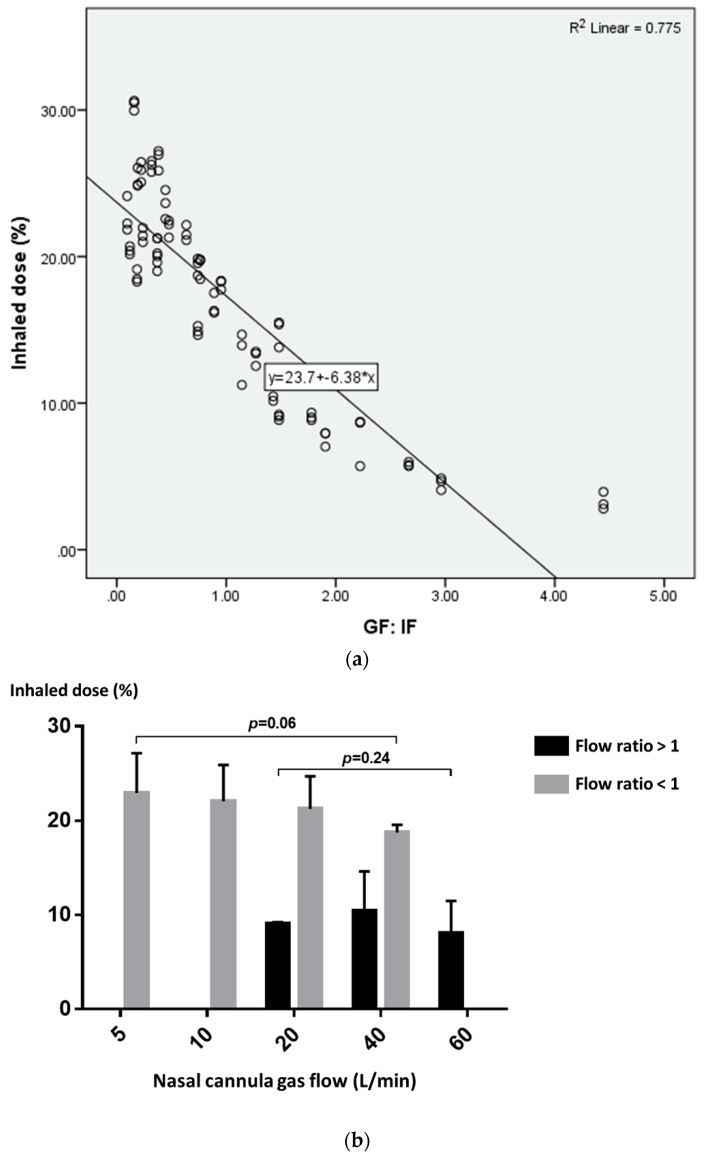
(**a**) Scatterplot of the inhaled dose and the GF:IF ratio; (**b**) inhaled dose with the ratio of GF:IF < 1 and > 1 in different nasal cannula flows.

**Figure 5 pharmaceutics-11-00225-f005:**
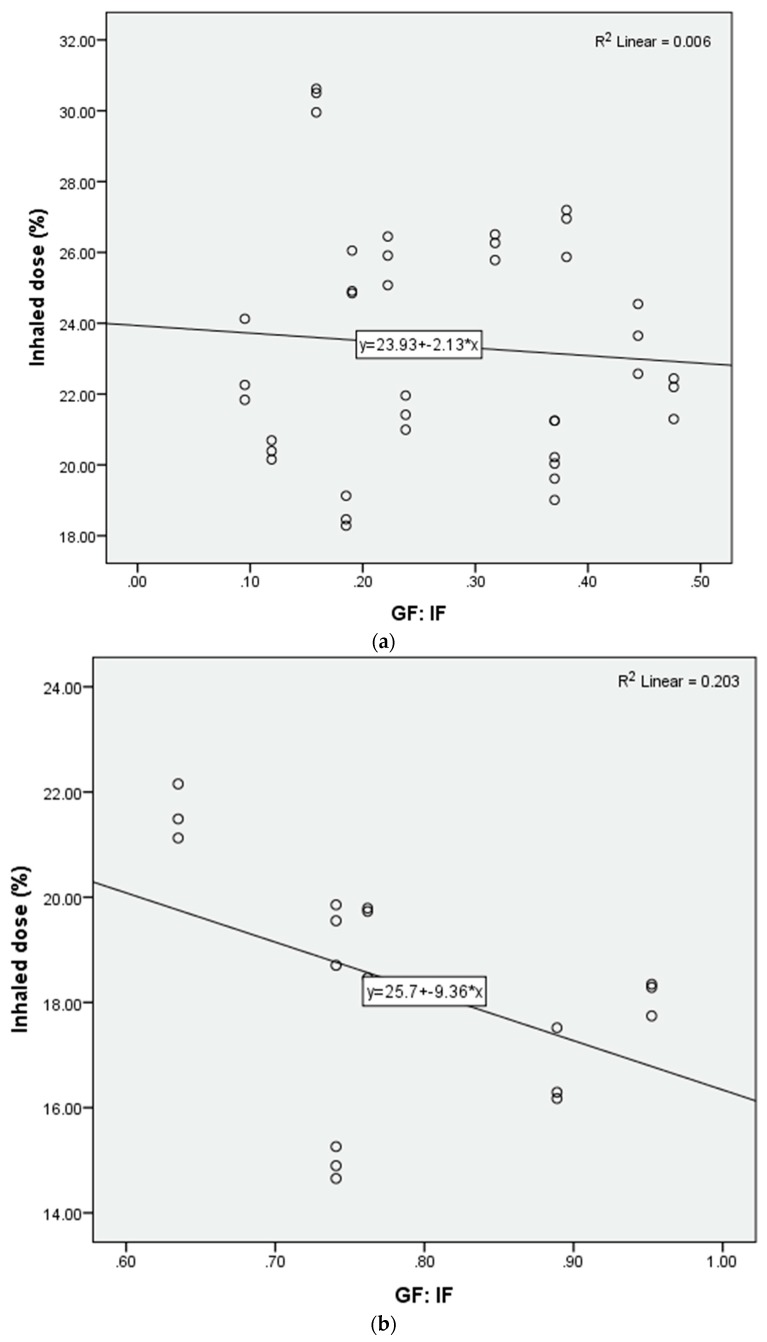
Scatterplot of inhaled dose and different GF: IF ratios: (**a**) 0.1–1.0; (**b**) 0.1–0.5; (**c**) 0.51–1.0.

**Table 1 pharmaceutics-11-00225-t001:** Breathing profiles in experiments.

Breathing Pattern	*Vt* (mL)	*RR* (bpm)	*I:E*	*Ti* (s)	Aerosol Inhalation Time (*RR × Ti*) (s)	Inspiratory Flow (L/min)	Nasal Cannula Gas Flow (L/min)	Ratio of Nasal Cannula Gas Flow to Patient Inspiratory Flow
Quiet breathing	300	15	1: 2	1.33	20	13.5	5, 10, 20,40,60	0.37, 0.74, 1.48, 2.96, 4.44
	500	15	1: 2	1.33	20	22.5	5, 10, 20,40,60	0.22, 0.44, 0.88, 1.78, 2.67
	700	15	1: 2	1.33	20	31.5	5, 10, 20,40,60	0.16, 0.32, 0.64, 1.27, 1.90
Distressed breathing	450	30	1: 1	1	30	27	5, 10, 20,40,60	0.19, 0.37, 0.74, 1.48, 2.22
	700	30	1: 1	1	30	42	5, 10, 20,40,60	0.12, 0.24, 0.48, 0.95, 1.43
	700	30	1:1.5	0.8	24	52.5	5, 10, 20,40,60	0.10, 0.19, 0.38, 0.76, 1.14

*Vt*, tidal volume; *RR*, respiratory rate; *I:E*, inspiratory to expiratory time ratio; *Ti*, inspiratory time.

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
