# Peer review of "The Ratio of Nasal Cannula Gas Flow to Patient Inspiratory Flow on Trans-nasal Pulmonary Aerosol Delivery for Adults: An in Vitro Study"

_pharmaceutics, 2019, doi:10.3390/pharmaceutics11050225_

Round 1

Reviewer 1 Report

The manuscript presents the effect of ratio of nasal cannula gas flow to patient inspiratory flow on aerosol delivery to lung through nose. The conclusion of the study that having a lower gas cannula flow than inspiratory flow results in higher drug delivery, has high clinical implications. The manuscript is well written and results are analyzed in detail.

The following are some of the comments which may help to improve the manuscript.

 Even though the study results show that the low cannula gas flow results in increased trans-nasal drug delivery, was there any substantial increase in aerosol deposition in the nasal cannula and the associated circuits. I would assume that the decrease in cannula flow would result in increased aerosol residence time. In a humidified condition, this would result in hygroscopic growth. These aerosols with increased size can deposit in the cannula and the connecting circuit before reaching the nasal cavity. As pointed out in previous studies, continuous deposition together with humidified conditions can cause flow obstruction and liquid sputtering. Can the authors comment on it.

error in page 7 line 231 - "air entertainment"

Author Response

Even though the study results show that the low cannula gas flow results in increased trans-nasal drug delivery, was there any substantial increase in aerosol deposition in the nasal cannula and the associated circuits. I would assume that the decrease in cannula flow would result in increased aerosol residence time. In a humidified condition, this would result in hygroscopic growth. These aerosols with increased size can deposit in the cannula and the connecting circuit before reaching the nasal cavity. As pointed out in previous studies, continuous deposition together with humidified conditions can cause flow obstruction and liquid sputtering. Can the authors comment on it.

We didn’t study the amount of aerosol deposited in the nasal cannula and circuits. We agree that the aerosol deposition in the nasal cannula and circuits might increase, because of the hygroscopic growth of the particle size. However, when the administered gas flow is higher than patient inspiratory flow, the impact of flow rate on aerosol deposition might be higher than the impact of aerosol hygroscopic growth, as a significant amount of aerosol is wasted. In contrast, when the gas flow is lower than patient inspiratory flow, the impact of hygroscopic growth might become important. This might explain the similarity of aerosol deposition with the flow ratio of 0.1-0.5 during trans-nasal aerosol delivery in our findings. Longer residence time with sedimentation is more likely the cause for difference than extended exposure to high absolute humidity, as hygroscopic changes occur within seconds. This impact is added in discussion section accordingly.

2. Error in page 7 line 231 - "air entertainment": Done

Reviewer 2 Report

The study is well designed and the manuscript is well written and included the study limitations towards the end of the manuscript. I have no reservations or further comments.

Author Response

We thank reviewers' encouragement. No comment.

Reviewer 3 Report

The manuscript entitled “The Ratio of Nasal Cannula Gas Flow to Patient Inspiratory Flow on Trans-nasal Pulmonary Aerosol Delivery for Adults: An in vitro Study”, deals with the influence of gas flow on aerosolized medication in patients with oxygen administration. This topic is of high interest in the clinical practice since many patients receiving oxygen supply suffer from pulmonary illness that may benefit from drug delivery by nasal inhalation. The aim and hypothesis of the study are well defined. This is a carefully executed study and the results are well presented supporting the conclusions achieved.

Major point

Despite the experimental work being well designed and according to the aims, there is a weak point related to the assumption of the collecting filter being an “absolute filter. According to literature data (Respiratory Care, 2016, vol 61, pg 1710-11) Respirgard II 303 does not always behave as an absolute filter. The authors must check this for Albuterol in order to make sure that the filter does not allow the drug to go across. This has to be checked for low and high gas flows.

Minor points

Pg 2, lines 58-60. This sentence is not clear. Ari speculates ………..  What does Ari mean?

Pg 4 lines 107-111. This paragraph should be in the previous section (Experiment set up).  This deals with a very relevant issue influencing the discharged dose. A more detailed explanation of this experimental procedure should be presented, for example nebulization time, nebulizer connection device to the nasal cannula, etc

Author Response

Despite the experimental work being well designed and according to the aims, there is a weak point related to the assumption of the collecting filter being an “absolute filter. According to literature data (Respiratory Care, 2016, vol 61, pg 1710-11) Respirgard II 303 does not always behave as an absolute filter. The authors must check this for Albuterol in order to make sure that the filter does not allow the drug to go across. This has to be checked for low and high gas flows.

We take the reviewers point, but we do not assume the Respirgard II 303 is an absolute filter. However that filter has been commonly used in aerosol bench studies with extensive validation at both high and low flows, demonstrating that the Respirgard filter captures > 98% of medical aerosol in the 0.5 – 5 micron range. Moreover, as the author (Dr. Walsh) stated in the study (Respiratory Care, 2016, vol 61, pg 1710-11), “no absolute filter is 100% efficient”. The Respirgard II 303 filter is rated at 99.9% bacterial and 99.8% viral efficiency of 0.3 µm or larger-sized particles as measured by the Nelson Laboratory, Inc. this means >99% of respirable (0.5 - 3 µm) could be captured by the Respirgard II 303 filter. The aerosol penetrating the filter is primarily in the small particle size (≤ 0.3 µm), which represents such a small mass of the emitted aerosol that it has little or no meaning on clinical practice, and that a large proportion of these  aerosols would not be deposit on the lower airways and would be exhaled by patients.

2. Pg 2, lines 58-60. This sentence is not clear. Ari speculates ………..  What does Ari mean?

Dr. Ari is a researcher who has implemented a lot of aerosol studies and published significant amount of papers on aerosol studies, especially on aerosol inhalation via high-flow nasal cannula. In her recent published review, she speculates the aerosol deposition might be optimized when the gas flow matches patient inspiratory flow. It is changed to “Dr. Ari in her recently published review”, in order to avoid confusion.

3. Pg 4 lines 107-111. This paragraph should be in the previous section (Experiment set up).  This deals with a very relevant issue influencing the discharged dose. A more detailed explanation of this experimental procedure should be presented, for example nebulization time, nebulizer connection device to the nasal cannula, etc

We didn’t record the nebulization, but a standard volume (1mL) was utilized in each run. “Nebulization took 2-4 minutes to complete” is added.

The aerogen nebulizer was placed at the inlet of humidifier, per manufacture’s recommendation, which is also a standard and common clinical practice. “A mesh vibrating nebulizer (Aerogen Solo, Aerogen Ltd., Ireland) with a residual drug volume <0.1mL was placed via a T-piece at the dry side (inlet) of the humidifier” is added.